# Searching for the Role of the *IFNγ* rs2430561 Polymorphism in Inducible Inflammation: Contribution to Metabolic Syndrome in 45 to 60-Year-Old Women

**DOI:** 10.3390/ijerph16050884

**Published:** 2019-03-11

**Authors:** Małgorzata Szkup, Aleksander Jerzy Owczarek, Anna Lubkowska, Elżbieta Chełmecka, Karolina Skonieczna-Żydecka, Elżbieta Grochans

**Affiliations:** 1Department of Nursing, Pomeranian Medical University in Szczecin, Żołnierska Str. 48, 71-210 Szczecin, Poland; szkup.m1@gmail.com; 2Department of Statistics, Department of Instrumental Analysis, School of Pharmacy with Division of Laboratory Medicine in Sosnowiec, Medical University of Silesia in Katowice, Ostrogórska Str. 30, 41-200 Sosnowiec, Poland; aowczarek@sum.edu.pl (A.J.O.); echelmecka@sum.edu.pl (E.C.); 3Department of Functional Diagnostics and Physical Medicine, Pomeranian Medical University in Szczecin, Żołnierska Str. 54, 71-210 Szczecin, Poland; annalubkowska@gmail.com; 4Department of Biochemistry and Human Nutrition, Pomeranian Medical University in Szczecin, Broniewskiego Str. 24, 71-460 Szczecin, Poland; karzyd@pum.edu.pl

**Keywords:** metabolic syndrome, IFNγ polymorphism, tryptophan-kynurenine pathway, chronic inflammation, women’s health

## Abstract

Metabolic syndrome (MetS) is a cluster of conditions, increasing the risk of developing diseases that can lead to premature death. Interferon γ-inducible (the production of which is dependent on the *IFNγ* rs2430561 polymorphism) tryptophan-kynurenine inflammatory cascade helps to understand the increased association between inflammatory process and MetS, which is why we seek the relationship between the *IFNγ* gene polymorphisms and serum levels of markers of interferon-gamma (IFNγ)-inducible inflammatory cascade. The study sample consisted of 416 women, including 118 (28.4%) with MetS. The research procedure involved interview, anthropometric measurements, and blood collection. Kynurenine levels were significantly higher in the group of women with MetS. In the group with MetS, the A/T genotype of the *IFNγ* gene was accompanied by higher kynurenine levels. A direct relationship between the *IFNγ* gene polymorphisms and the rest of the markers of IFNγ-inducible inflammatory cascade was not confirmed with regard to MetS in 45 to 60-year-old women. A disparity in the kynurenine level, as well as the relationship between the presence of the A/T genotype of the *IFNγ* gene and a higher level of kynurenine in the group of women with MetS, may indicate an association between inflammation, metabolic disorders and tryptophan-kynurenine inflammatory cascade.

## 1. Introduction

An increase in the average lifespan means that currently as many as 30% of a woman’s life falls within the postmenopausal period. Therefore, health problems faced by postmenopausal women have become a tremendous challenge for modern medicine [1]. Metabolic syndrome (MetS) is defined as a cluster of metabolic abnormalities, enhancing the risk of cardiovascular disease and premature death. According to estimates, MetS affects 20 to 30% of the middle-aged population [2,3]. The incidence of such disorders increases with age [4,5]. The risk of MetS has been found to be significantly higher in the postmenopausal period, regardless of age and commonly known risk factors of cardiovascular disease [6,7]. Aside from environmental impact, genetic and inflammatory issues are also believed to play an important role in their development [8]. 

Although currently, inflammatory processes are not included in the definition of MetS, a chronic low-grade inflammatory state is regarded as one of the causes of MetS [2,5]. Interferon-gamma (IFNγ), which is the key Th-1 type pro-inflammatory cytokine, transcriptionally induces the rate-limiting enzyme of the tryptophan (TRY)–kynurenine (KYN) pathway, indoleamine 2,3-dioxygenase (IDO). Activation of IDO shunts TRY metabolism from the production of serotonin (5-HT) towards the production of KYN and its derivatives [9]. 

The production of IFNγ is dependent on the polymorphic *IFNγ* rs2430561 gene, which possibly includes the T allele contributing to the high IFNγ production, and the A allele responsible for the low IFNγ production, located at the position +874 of the intron 1 of *IFNγ* gene (+874 IFNG). Average levels of IFNγ cytokines are higher in healthy carriers of the T allele than in those with the A allele. The study of Pravic et al. shows clearly that the T to A polymorphism can directly influence the level of IFNγ production [10]. IFNγ induces IDO activity, therefore, the T allele carriers are probably characterized by a higher IDO activity than those with the A allele. This hypothesis has been supported by the research conducted among healthy individuals in Finland. It demonstrated that the presence of the high-production T allele was associated with an increased IDO activity. However, this was only observed in females [11]. The production of IFNγ in women over 40 and in the postmenopausal period is considerably higher than in younger patients [12].

Thus, TRY-KYN metabolism may be genetically determined through regulatory effects of genes on IDO activity [13]. 

### TRY Metabolism Pathways 

TRY is an essential amino acid with two non-protein metabolic pathways: methoxyindole and KYN. In the first pathway, 5-HT and melatonin are formed. The availability of TRY as a substrate is one of the rate-limiting factors of 5-HT biosynthesis since less than 5% of TRY is metabolized along this pathway [14]. 5-HT, on the other hand, is a substrate for melatonin synthesis [15]. Neurotropic activity of KYNs suggests that up-regulation of the TRY-KYN pathway contributes to 5-HT deficiency, resulting in the acceleration of depression-associated anxiety, psychosis, and cognitive decline [16]. In the second pathway, about 95% of TRY is metabolized and converted into KYN, i.e., metabolite having adverse effects on the central nervous system [14,17]. 

The rate-limiting enzymes of KYN formation from TRY are IDO and TRY 2,3-dioxygenase (TDO) [17,18]. IDO catalyzes the same biochemical reaction as TDO does simultaneously in physiological conditions, whereas in pathological states, IDO activity suddenly increases, and TDO decreases. IDO activity depends on the availability of TRY, hormones, and cytokines. An increase in the IDO activity is induced by proinflamatory cytokines [19], while the influence of environmental stress factors is a mediator in hormonal activation of TDO [16]. 

KYN inhibits TRY transport via the blood–brain barrier, stimulates IDO activity, and causes anxiety states, which was observed in Lapin’s study performed on an animal model (mice) [20]. KYN is metabolized in one of two pathways: the first of them leads to the formation of kynurenic acid (KYNA), and the second—nicotinamide adenine dinucleotide (NAD) [13]. KYNA seems to play a part in the development of cognitive disorders observed in depression, schizophrenia, and dementia. Moreover, it is associated with the occurrence of obesity and hypertension [21,22]. At the same time, it may exert anti-inflammatory effects through inhibiting the lipopolysaccharide-induced secretion of tumor necrosis factor-alpha (TNF-alpha) in the mononuclear cells of peripheral blood [23]. 

The second KYN metabolic pathway leads to the formation of nicotinamide adenine dinucleotide (NAD) as a final product, which plays a major role as a coenzyme in numerous oxidation–reduction reactions. Over 30 indirect metabolites include 3-hydroxykynurenine and 3-hydroxyanthranilic acids [14,15]. 3-hydroxykynurenine is a neurotoxin that is probably involved in the development of neurodegenerative diseases [17]. Furthermore, KYN, 3-hydroxykynurenine, and 3-hydroxyanthranilic acids take part in the induction of apoptosis [24]. Xanthurenic acid forms a complex with insulin, which causes a decline in sensitivity to insulin and decreases its activity, thus, leading to insulin resistance typical of MetS [24]. What is more, all acids mentioned above increase the production of inflammatory factors, such as prostaglandin and leukotrienes [15,25]. Presumably, this can trigger a mechanism which, in combination with progressing age, may result in depression and atherosclerosis, as was underlined in a two-stage study (baseline and 3-year follow-up) performed on a group of 324 individuals [26]. 

The KYN pathway can be modulated by both endogenous and exogenous factors. The regulation of the KYN-NAD metabolic pathway may be influenced by hormones and cytokines. It is assumed that KYNs can influence the process of aging and the age of menopause. Therefore, it is worth emphasizing that estradiol and progesterone cause a significant decline in the level of kynurenine 3-mono-oxydase—a key enzyme of the KYN-NAD metabolic pathway, which was confirmed by the analysis of the macaque dorsal raphe region [27]. Estrogens inhibit another important enzyme—kynureninase—in postmenopausal women [28]. The enzyme rate in the TRY-KYN metabolic pathway may be regulated by hormones, cytokines, and IDO [13].

IDO is an enzyme induced by cytokines, such as IFNγ, Interferon-alpha, TNF-alpha, and Interleukin-1, Interleukin-12, Interleukin-18, and prostaglandin E2. The influence that has been best documented so far is that of IFNγ-induction of IDO [29].

The hypothesis of the IFNγ inducible TRY-KYN inflammatory cascade helps to understand the increased association between aging, inflammation, and metabolic disorders.

The purpose of this study was to seek the relationship between the *IFNγ* gene polymorphism and serum levels of markers of the IFNγ-inducible inflammatory cascade in 45 to 60-year-old women with MetS. 

## 2. Materials and Methods

The investigation was conducted in compliance with ethical standards, the Declaration of Helsinki, and national and international guidelines. The protocol of the study was approved by the Bioethical Commission of the Pomeranian Medical University of Szczecin, Poland (approval number KB-0012/181/13). The participants’ informed written consent has been obtained.

The study included patients from the general population of West Pomeranian Province (Poland). The whole study sample consisted of 416 women, including 118 (28.4%) with MetS. The inclusion criteria were: female sex, 45 to 60 years of age (mean with SD: 53 ± 5 years), the lack of current inflammatory, psychiatric or cancerous diseases, and informed written consent to take part in the study. Thirty-three point six percent of the respondents had primary education, 27.7% had tertiary education, 26.9% had secondary education, and 11.8% had vocational education. Forty-four point eight percent of the patients lived in big cities (more than 100,000 residents), 39.4% lived in rural areas, and the remainder lived in smaller cities. Eighty-two point seven percent of the women were married, 6.2% cohabited with their partners, and 11.6% were single.

The research procedure consisted of three stages: interview, anthropometric measurements, and blood collection (for biochemical analysis, genetic analysis, and the measurement of inflammation markers’ levels).

1. Interview: we collected basic sociodemographic data, as well as information concerning pharmacotherapy for hypertension, hypertriglyceridemia, hyperglycemia, and low high-density lipoprotein (HDL) levels. The patients were also asked about their current inflammatory, psychiatric, and cancerous diseases;

2. Anthropometric measurements: blood pressure was gauged according to the procedure by registered nurses. The waist was measured in a standing position between the lower rib margin and the upper margin of the iliac crest at the end of a gentle exhalation;

3. Blood collection: venous blood was collected according to relevant rules and procedures concerning collecting, storing, and transporting biological material from the antecubital vein, between 7.00 and 9.30 in the morning after overnight fasting and a 10 min rest in a sitting position. We collected blood separately into two Vacutainer tubes (Sarstedt, Nümbrecht, Germany): one with 1 g/L K2, Ethane-1,2-diyldinitrilotetraacetic acid and the other one for biochemical analysis of serum (7 mL). The levels of fasting glycemia, triglycerides (TG) and HDL were determined. Next, DNA was isolated for genetic analysis of the *IFNγ* rs2430561 polymorphisms. The rest of the blood was used to assess the levels of inflammatory markers (IFNγ, TRY, KYN, IDO, 5-HT). 

A subgroup meeting the International Diabetes Federation (IDF) diagnostic criteria for MetS from 2009 [30] was selected from all the women surveyed and marked as MetS+. The remaining patients were allotted to the second group without MetS (MetS−). Patients were included in the MetS+ group if they had at least three out of five symptoms: waist size ≥80 cm; fasting glycemia ≥100 mg/dL (5.6 mmol/L) or related pharmacotherapy; TG level ≥150 mg/dL (1.7 mmol/L) or related pharmacotherapy; HDL cholesterol level ≤50 mg/dL (1.3 mmol/L) or related pharmacotherapy; blood pressure: systolic blood pressure ≥130 and/or diastolic blood pressure ≥85 mmHg or related pharmacotherapy. 

The research results presented in this manuscript are a part of a larger project, based on the same group, titled “The search for factors contributing to the metabolic syndrome among women aged 45–60 years”, in which we also provided more detailed information about the description of the research procedure [31]. 

### 2.1. DNA Isolation and the IFNγ rs2430561 Gene Polymorphism Genotyping

Genomic DNA was isolated from the whole blood according to standard salting procedures [32]. 

A polymorphism in the *IFNγ* was genotyped with the Real-Time PCR using the Light Cycler II. The LightSNiP primers and probe for rs1800629 were used in the assay (TIB MOLBIOL GmbH, Berlin, Germany). Reaction conditions and concentrations of reagents in the reaction mixture were in accordance with manufacturer’s instructions. The amount of DNA in the sample was about 50 ng for a total sample volume of 20 µL.

### 2.2. The Measurement of Markers of the IFNγ-Inducible Inflammatory Cascade

The serum levels of IFNγ, TRY, KYN, IDO, and 5-HT were measured by immune-enzymatic assays using commercially available enzyme-linked immunosorbent (ELISA) kits according to the manufacturer’s protocol. IFNγ [IU/mL] with DRG (Marburg, Germany) with 0.03 IU/mL limit detection; intra-assay was 3.2% and inter-assay 5.8%. For TRY [µg/mL] we used LDN (Nordhorn, Germany) with 1.2 µg/mL limit detection; intra-assay was 11% and inter-assay 15%. KYN [ng/mL] Labor Diagnostika Nord (Nordhorn, Germany); 45.7 ng/mL limit detection; intra-assay was 12.9% and inter-assay 17.5%. IDO [ng/ml] Sunred, Shanghai with limit detection 0.238 ng/mL; intra-assay was <10% and interassay <12%. 5-HT [ng/mL] LDN (Nordhorn, Germany) with limit detection 6.2 ng/mL; intra-assay was 9.7% and inter-assay 12.4%.

### 2.3. Statistical Analysis 

Statistical analysis was performed using Statistica 13 PL (TIBCO, Palo Alto, CA, USA) and R (CRAN) software. Statistical significance was set at *p* < 0.05. All tests were two-tailed. No data imputation has been done. Interval data were expressed as a mean ± standard deviation in the case of normal distribution, and as a median/lower–upper quartile range in the case of data with skewed or non-normal distribution. The distribution of variables was evaluated by the Shapiro–Wilk test and the quantile–quantile plot. The homogeneity of variance was assessed by the Fisher–Snedecor test. In the case of skewed data distribution, logarithmic transformation was done before analysis. The following tests were used to verify hypotheses: the parametric test for two independent samples (Student’s *t*-test) in the case of normal distribution or after logarithmic transformation, and the Mann–Whitney U test if the distribution was not normal. Nominal and ordinal data were compared with the χ^2^ test. The analysis of the association between the *IFNγ* gene rs2430561 polymorphisms and the levels of IFNγ-inducible inflammatory cascade markers were done on the basis of generalized linear models (for quantitative traits—quantitative trait loci (QTL) analysis) with *SNPassoc* package in R software (R version 3.5.1, R Core Team (2013). R: A language and environment for statistical computing. R Foundation for Statistical Computing, Vienna, Austria). The co-dominant model was used, and the model yields sample size, mean, and standard error for each genotype, as well as mean difference and its 95% confidence interval [33]. There were no multiple comparisons problem as each trait was analyzed independently to another.

## 3. Results

We compared the women qualified for subgroups MetS+ and MetS−. All parameters statistically significantly differed between both groups (*p* < 0.001). Particular attention should be paid to the fact that even in the MetS− group, the average values for one of the five symptoms—waist size-analyzed exceeded the values considered by the IDF as correct. The mean values for three out of five symptoms in both groups were within normal ranges: TG level, HDL level, and diastolic blood pressure. In the MetS+ group, the average values of other MetS parameters were elevated, while in the MetS− group they were normal: fasting glycemia and systolic blood pressure. Accurate characteristics of the study sample with regard to division into MetS+ and MetS− groups, according to IDF diagnostic criteria from 2009, is available at Szkup et al. because the research results presented in this manuscript are a part of a larger project, based on the same group [31].

Analysis of IFNγ-inducible inflammatory cascade markers did not demonstrate any statistically significant differences between both groups. The only exception was the level of kynurenine, which was significantly higher in the MetS+ group (Table 1).

The T/T genotype was the most frequent of the *IFNγ* gene variants (45%). The distribution of genotypes and alleles in both analyzed groups was very similar. There were no statistically significant differences between the patients (Table 2).

As the last stage of the study, we tested the association between the *IFNγ* gene rs2430561 polymorphism and inflammatory pathway markers with regard to MetS. The levels of IFNγ, TRY, IDO, and 5-HT were not statistically significantly related to any of the tested *IFNγ* genotypes in any of the MetS groups. In the MetS+ group, the A/T genotype of the *IFNγ* gene was accompanied by higher KYN levels than in the MetS− group (Table 3). 

## 4. Discussion

The IFNγ-inducible TRY-KYN inflammatory cascade helps to understand the association between inflammation and metabolic disorders, which is now recognized as a crucial mechanism in the development of MetS. IFNγ production is determined by the *IFNγ* rs2430561 polymorphism, which is why we had decided to assess the relationship between the polymorphisms of this gene and serum levels of IFNγ-inducible inflammatory cascade markers. Some researchers have focused on the relationship between polymorphic gene variants and particular symptoms typical of MetS. However, there are no reports on the relationship between serum levels of IFNγ-inducible inflammatory cascade markers and the *IFNγ* rs2430561 polymorphism in patients with MetS.

### 4.1. Markers of the IFNγ-Inducible Inflammatory Cascade

Recent studies have shown that IFNγ may play a crucial role in obesity-related inflammatory response. White adipose tissue secretes IFNγ, which induces proinflammatory cytokines, such as Interleukin-1, Interleukin-6 and TNF-alpha [34]. This relationship has been confirmed by the research on mice. It was observed that the production of IFNγ in obese mice was higher than in the control group, and accompanied by IFNγ receptor deficiency [35]. IFNγ participates in the activation of serotonin transporters, which leads to a decline in its level in the extracellular space [36]. Cytokines influence 5-HT levels, and also through their impact on the metabolism of KYN pathway enzymes, for example, IFNγ increases the activity of IDO, and, thus, by shifting TNF from the 5-HT production pathway and by reducing its level, increase the levels of TNF metabolites, which have adverse effects on the central nervous system [37]. In our study, we failed to confirm the difference in the levels of IFNγ between the patients with and without Mets.

The study conducted by Niinisalo et al. indicates a significant relationship between IDO activity and risk factors of atherosclerosis, such as age, carotid artery intima-media thickness, low density lipoprotein concentration, and body mass index [38]. According to Pertovaara, IDO enzyme is involved in the immune regulation of early atherosclerosis and can be a novel marker of immune activation in early atherosclerosis in females [39]. The research on a group of people aged between 21 and 64 (mean age 45) demonstrated that IDO activity increases with age [11]. The prospective study provided evidence that high initial IDO activity may be a predictive factor of mortality in the elderly [40]. Our analysis did not confirm the existence of differences in IDO levels between the groups of women with and without MetS. 

The lower KYN to TRY ratio is typical of patients with coronary heart disease. It has been noticed that in coronary heart disease, immune activation is associated with an increased rate of TRY degradation, resulting in its lower levels. The study mentioned above also provided information about the pathogenesis of mood disturbances and depression in patients with coronary heart disease [41]. Furthermore, it was noticed that free TRY levels in the blood serum of obese rats dropped, whereas the levels of other amino acids, which compete with TRY for transport across the blood–brain barrier, were elevated [42]. Lower TRY levels were also observed in obese patients, despite the reduction of body mass and changes in the diet. It was additionally observed that TRY metabolic changes can subsequently reduce 5-HT production and cause mood disturbances, depression, and impaired satiety, ultimately leading to increased caloric intake and obesity [43]. In the patients undergoing surgical treatment due to pathological obesity, the KYN to TRY ratio was significantly lower compared to the control group. Postoperative reduction of the body mass did not result in the normalization of the KYN/TRY ratio [44]. Our study did not demonstrate any difference in 5-HT levels between the groups with and without MetS. We noticed, however, that the level of KYN was statistically significantly higher in the women with than those without MetS. TRY levels were lower in the MetS+ group, but the difference was not statistically significant.

### 4.2. Relationships between the IFNγ Gene rs2430561 Polymorphism and the Levels of IFNγ-Inducible Inflammatory Cascade Markers with Regard to MetS 

The majority of available studies suggest that most symptoms of MetS are associated with a decline in IDO activity, an increased production of proinflammatory cytokines, especially IFNγ, and a higher incidence of highly productive alleles of the genes contributing to proinflammatory cytokines production [13]. 

The research conducted in populations of European ancestry shows that up-regulation of IFNγ production correlates with specific components of inflammation-associated MetS, total and disease-specific mortality, and aging [45,46]. In the study of Tsiavou et al., the frequency of the low-production A allele of the *IFNγ* rs2430561 gene was significantly higher in people with type 2 diabetes mellitus than in the controls [47]. Unlike in healthy participants [11], there was no relationship between the presence of the high-production T allele and high *IFNγ* production in patients with a diagnosis of type 2 diabetes mellitus. This was probably caused by the activation of the molecular inhibition of IFNγ production [47]. While some studies provide evidence for the association between this *IFNγ* polymorphism and diabetic retinopathy (the Indian ethnic group) [48] and with neuropathy (the South Indian patients) [49], others (Brazilian individuals) do not confirm its existence [50]. Rodrigues et al. did not observe any relationship between the *IFNγ* polymorphisms and the occurrence of hypertension, dyslipidemia, and obesity in patients with type 2 diabetes. They found, however, that the A/A genotype was accompanied by lower glucose levels, and concluded that the decreased expression of IFNγ may contribute to downregulation of inflammatory response in patients with type 2 diabetes, which enables better glycemic control [48]. The results obtained by Garg et al. suggest a significant role of the *IFNγ* T allele in coronary heart disease due to its direct effect on diastolic hypertension, but the relationship between polymorphic forms of this gene and diabetes, body mass index, serum triglycerides or very low density lipoproteins has not been confirmed. Despite this, the T allele was considered a strong determinant of coronary heart diseases in the Indian population [51].

A consequence of up-regulation of IFNγ production is an elevation of neopterin, the concentration of which correlates with abdominal obesity, HDL cholesterol, and insulin resistance. The research carried out by Oxegen shows that neopterin correlates with clinical markers of MetS and mortality risk in the population with a different genetic background, i.e., adult Boston community dwellers of Puerto Rican origin [52]. The novelty of our study is that it analyzes the relationship between the *IFNγ* gene polymorphism and MetS. To the best of our knowledge, such studies have not yet been conducted.

The studies of humans have shown that the T (high production) allele (T/A + T/T genotypes) contributes to higher expression of IFNγ mRNA [53] and higher blood levels of IFNγ than A/A genotypes [54]. The study conducted among healthy women in Finland indicates that TRY-KYN metabolism is controlled by the *IFNγ* gene via its influence on the regulation of IDO activity. The T high production allele was associated with increased IDO activity compared with the A allele. The influence of the *IFNγ* gene polymorphism on the level of TRY in healthy subjects was assessed. The authors found that the women with the genotype T/T had the highest TRY and KYN concentrations. This effect was observed in KYN/TRY ratios as well. In the men, this effect was not observed [11]. Centenarians are characterized by a higher frequency of genetic markers associated with better control of inflammation, including a higher frequency of A alleles of *IFNγ* gene which encodes the production of IFNγ protein [55]. In our study, we confirmed the existence of a relationship between carriers of the A/T genotype and KYN level in the MetS+ group, but no relationship was found between the *IFNγ* polymorphisms and the levels of IFNγ, TRY, KYN, IDO, and 5-HT with regard to MetS. The distribution of the genotypes and alleles was very similar in both groups, which might have contributed to the lack of a direct relationship between the *IFNγ* rs2430561 polymorphism and MetS.

Some limitations should be mentioned with the hope to stimulate further evaluation of the proposed hypothesis. In our study, the patients were qualified to the MetS+ group on the basis of the IDF criteria from 2009. Nevertheless, the symptoms included in the MetS definition have changed many times over time, including elements, such as hyperlipoproteinemia, hyperuricemia [56], insulin resistance [57,58], endothelial dysfunction, and microalbuminuria [59]. Evaluation of all components of MetS in daily clinical practice would be very difficult. Therefore, we decided that the diagnosis of MetS must be based on unambiguous, reliable, and simple rules [59]. The most rigorous criterion adopted in our study was waist circumference, which according to the International Diabetes Federation should be less than 80 cm in women (according to the Third Report of the National Cholesterol Education Program, Expert Panel on Detection, Evaluation, and Treatment of High Blood Cholesterol in Adults Treatment Panel III it should not exceed 88 cm). Another methodology, and, thus, different criteria for including patients in the MetS+ group, could lead to different results than those presented in this paper. More than that, because of a preliminary character of the study of the Polish population, we did not use multiple testing corrections. Therefore, the results should be interpreted with caution.

## 5. Conclusions

A direct relationship between the *IFNγ* gene polymorphisms and the markers of IFNγ-inducible inflammatory cascade with regard to MetS in 45 to 60-year-old women was not confirmed. However, a disparity in the KYN level, as well as the relationship between the carrier of the A/T genotype of the *IFNγ* gene and the higher level of KYN in the group of women with MetS, may indicate a significant association between inflammation, metabolic disorders, and TRY-KYN inflammatory cascade.

## Figures and Tables

**Table 1 ijerph-16-00884-t001:** Characteristics of interferon-gamma (IFNγ)-inducible inflammatory cascade with regard to metabolic syndrome (MetS).

Inflammation Markers	MetS+ *N* = 118	MetS− *N* = 298	*p*
IFNγ [IU/mL]	0.04 (0.03–0.21)	0.05 (0.03–0.14)	0.936
Tryptophane [µg/mL]	12.63 (9.43–15.64)	12.76 (9.46–15.78)	0.643
Kynurenine [ng/mL]	680.2 (483.3–824.1)	607.7 (490.2–731.1)	<0.05
Indoleamine [ng/mL]	14.98 (11.50–45.92)	20.49 (12.88–44.74)	0.166
Serotonin [ng/mL]	116.6 (75.8–171.7)	120.3 (80.4–193.0)	0.489

Median (lower quartile–upper quartile); *p*—significance level in the *t*-Student test or in the Mann–Whitney U test; IFNγ—interferon γ.

**Table 2 ijerph-16-00884-t002:** Analysis of the distribution of the *IFNγ* gene rs2430561 polymorphism with regard to MetS.

MetS	*IFNγ* Genotype			*IFNγ* Allele	
	T/T *n* (%)	A/T *n* (%)	A/A *n* (%)	T Allele *n* (%)	A Allele *n* (%)
MetS+	56 (47)	41 (35)	21 (18)	153 (65)	83 (35)
MetS−	132 (44)	120 (40)	46 (16)	384 (64)	212 (36)
Σ	188 (45)	161 (39)	67 (16)	537 (64)	295 (36)
*p*	0.564			0.913	

*n*—number of cases; Σ—sum of cases; *p*—significance level in the χ^2^ test.

**Table 3 ijerph-16-00884-t003:** Analysis of the relationships between the *IFNγ* gene rs2430561 polymorphism and the levels of IFN*γ*-inducible inflammatory cascade markers with regard to MetS.

*IFNγ* Genotype	MetS+	MetS−	Δ	± 95% CI
	Mean Value (SE); *N*	Mean Value (SE); *N*		
log_10_ (IFNγ [IU/mL])				
T/T	−1.049 (0.156); 56	−1.187 (0.543); 132	−0.138	−0.124–0.400
A/T	−0.951 (0.170); 41	−1.077 (0.062); 120	0.126	−0.152–0.404
A/A	−1.246 (0.126); 21	−1.040 (0.115); 46	−0.206	−0.626–0.214
log_10_ (Tryptophan [µg/mL])				
T/T	12.51 (0.531); 56	12.36 (0.455); 132	0.151	−1.665–1.968
A/T	12.21 (0.736); 41	13.81 (0.618); 120	−1.592	−3.652–0.468
A/A	15.56 (1.600); 21	13.92 (0.977); 46	1.642	−1.357–4.641
log_10_ (Kynurenine [ng/mL])				
T/T	2.771 (0.034); 56	2.750 (0.194); 132	0.021	−0.045–0.086
A/T *	2.833 (0.031); 41	2.743 (0.170); 120	0.090	0.015–0.164
A/A	2.822 (0.035); 21	2.758 (0.030); 46	0.064	−0.044–0.172
log_10_ (Indoleamine [ng/mL])				
T/T	32.43 (3.494); 56	31.17 (2.025); 132	1.266	−6.111–8.643
A/T	27.21 (3.702); 41	30.28 (2.140); 120	−3.071	−11.44–5.298
A/A	29.21 (5.132); 21	30.19 (3.184); 46	−0.977	−13.16–11.21
log_10_ (Serotonin [ng/mL])				
T/T	2.021 (0.046); 56	2.019 (0.031); 132	0.002	−0.110–0.113
A/T	2.068 (0.045); 41	2.115 (0.028); 120	−0.047	−0.174–0.079
A/A	2.084 (0.086); 21	2.132 (0.071); 46	−0.231	−0.232–0.135

*N*—number of cases; SE—standard error of the mean; Δ—mean difference between groups; ± 95% CI—95% confidence interval, IFNγ—interferon γ; *—statistically significant based on ± 95% CI; Co-dominant model for quantitative traits (QTL).

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
