# Peer review of "Searching for the Role of the IFNγ rs2430561 Polymorphism in Inducible Inflammation: Contribution to Metabolic Syndrome in 45 to 60-Year-Old Women"

_ijerph, 2019, doi:10.3390/ijerph16050884_

Round 1
Reviewer 1 Report
The role of the IFNγ rs2430561 polymorphism in inducible inflammation: contribution to metabolic syndrome in 45-60-year-old women
really think it is a well written papar, usually I have many coments regarding the methods used in the study, this time I really do not have much to say.
I have two small comments:
Table 4: I can not find it, or it is nowhere explained why the INFγ analysis is on fewer people than the analysis of other variables, especially that in Table 2 INFγ is analyzed on a whole group of patients ...
Trivia: Table 1: I am not convinced that it makes sense, because there is a comparison of 5 features in the MetS + and MetS- groups, but these are the features on the basis of which we qualify for these groups ...
Overall I really think it is a good paper.
Author Response
Dear Sir or Madam,
We are very grateful for the review of our article titled “The role of the IFNγ rs2430561 polymorphism in inducible inflammation: contribution to metabolic syndrome in 45-60-year-old women”. We would like to thank you for all your comments and suggestions, which helped us to improve our manuscript. We are grateful for good words about the general value of the manuscript, it is very motivating for us.
The following corrections have been introduced in order to address the suggestions of the Reviewer 1 (marked in the manuscript with blue):
1. Table 4: I can not find it, or it is nowhere explained why the INFγ analysis is on fewer people than the analysis of other variables, especially that in Table 2 INFγ is analyzed on a whole group of patients...
You are absolutely right, we had made a mistake during the preparation of the tables. We have improved it and changed the values in table 4:
IFNγ genotype | MetS+ | MetS- | D | ± 95% CI |
Mean value (SE); N | Mean value (SE); N | |||
log10 (IFNγ [IU/ml]) | ||||
T/T | -1.049 (0.156); 56 | -1.187 (0.543); 132 | -0.138 | -0.124 ¸0.400 |
A/T | -0.951 (0.170); 41 | -1.077 (0.062); 120 | 0.126 | -0.152¸0.404 |
A/A | -1.246 (0.126); 21 | -1.040 (0.115); 46 | -0.206 | -0.626¸0.214 |
2. Trivia: Table 1: I am not convinced that it makes sense, because there is a comparison of 5 features in the MetS + and MetS- groups, but these are the features on the basis of which we qualify for these groups...
Yes it's true. We placed this table to present the characteristics of the study sample. We think that it gives an interesting view of the health situation in both analyzed groups, allowing the reader to notice the most common problems, both among women with metabolic syndrome and healthy
We are very grateful for giving us the possibility of improving our manuscript. The article has been corrected according to the Reviewers’ suggestions.
Kindest regards,
Małgorzata Szkup
Reviewer 2 Report
The authors tried to elucidate the association between the IFNγ polymorphism and inflammatory trait of metabolic syndrome, and found that in the A/T genotype of the IFNγ gene, the MetS female group had higher level of kynurenine than the non-MeS.
“The distribution of the genotypes and alleles was very similar in both groups, which could contribute to the lack of a direct relationship between the TNF-alpha rs1800629 (IFNgamma?) polymorphism and MetS.” Line 313
The logic of the analysis is beyond my comprehension. Why did the authors analyze and discuss straight the relationship between the genotype and MetS or other single traits? Furthermore, the results indicates multiple comparison problem.
The Introduction should be revised to delete redundant explanation about nervous system.
The authors should follow the format of a usual academic table. Most vertical lines are unnecessary. The left end column of Table 4 has meaningless signs,””.
Lines 156-160. ‘The research results presented in this manuscript are a part of a larger project titled “The search for factors contributing to the metabolic syndrome among women aged 45-60 years”. We would like to encourage readers to familiarize themselves with the results, based on the same group, and regarding the influence of the TNF-alpha rs1800629 polymorphism on some inflammatory biomarkers [31],’ is awkward.
Line 120 “with” is italic?
Line 146 HDL or HDL-C?
Author Response
Dear Sir or Madam,
We are very grateful for the review of our article titled “The role of the IFNγ rs2430561 polymorphism in inducible inflammation: contribution to metabolic syndrome in 45-60-year-old women”. We would like to thank you for all your comments and suggestions, which helped us to improve our manuscript.
The following corrections have been introduced in order to address the suggestions of the Reviewer 1 (marked in the manuscript with grey):
1.“The distribution of the genotypes and alleles was very similar in both groups, which could contribute to the lack of a direct relationship between the TNF-alpha rs1800629 (IFNgamma?) polymorphism and MetS.” Line 313
Sorry, it was our mistake. We have changed to: “The distribution of the genotypes and alleles was very similar in both groups, which might have contributed to the lack of a direct relationship between the IFNγ rs2430561 polymorphism and MetS.”
2. The logic of the analysis is beyond my comprehension. Why did the authors analyze and discuss straight the relationship between the genotype and MetS or other single traits? Furthermore, the results indicates multiple comparison problem.
To solve this problem we have added the following sentence in the material and methods section: “There were no multiple comparisons problem as each trait was analyzed independently to another”. Additionally, in the limitation section, we have added The following information: “More than that, because of a preliminary character of the study in polish population, we did not use multiple testing corrections; therefore, the results should be interpreted with caution”.
3. The Introduction should be revised to delete redundant explanation about nervous system.
We have removed the sentence: “Transformation of TRY into 5-HT, and the latter into melatonin only takes place in the central nervous system”.
4. The authors should follow the format of a usual academic table. Most vertical lines are unnecessary. The left end column of Table 4 has meaningless signs,”¸”.
As suggested, we have changed sings ”¸”to “―” in table 4. We have removed two unnecessary lines but left the remaining lines, because in our opinion such arrangement of tables allows their easy and clear interpretation.
5. Lines 156-160. ‘The research results presented in this manuscript are a part of a larger project titled “The search for factors contributing to the metabolic syndrome among women aged 45-60 years”. We would like to encourage readers to familiarize themselves with the results, based on the same group, and regarding the influence of the TNF-alpha rs1800629 polymorphism on some inflammatory biomarkers [31],’ is awkward.
We did not mean advertising of our previous publication. We just wanted to draw the readers' attention to the fact that research on a different scope was already carried out on the same group of women. Because of the awkwardness we removed the fragment: “We would like to encourage readers to familiarize themselves with the results, based on the same group, and regarding the influence of the TNF-alpha rs1800629 polymorphism on some inflammatory biomarkers”.
6. Line 120 “with” is italic?
Corrected.
7. Line 146 HDL or HDL-C?
Throughout the article, we have consistently used the HDL shortcut for high-density lipoprotein.
We are very grateful for giving us the possibility of improving our manuscript. The article has been corrected according to the Reviewers’ suggestions.
Kindest regards
Małgorzata Szkup
Reviewer 3 Report
The study investigates the association between polymorphisms of the IFNγ gene and serum markers of inflammation in peri- and post-menopausal women. It seeks to establish possible causative factors for metabolic syndrome in this grouping. It is a carefully and clearly written manuscript which is appropriately referenced to convey its meaning and to give a good perspective on this area and set the current study in a wider context of other work in the field. It will be of interest to researchers in the area of inflammation and cytokine action and to those with an interest in public health.
There are some minor revisions to the writing style and quality of English usage that would improve clarity. For example, the authors should refine their use of the definite article. Moreover, there are a great many abbreviations used, not all of which are appropriately defined.
Abstract : l19 “whose production” should be changed to “the production of which”. L292 of the Discussion requires a similar revision.
Introduction :
L37 “causes” should be changed to “means”.
Paragraph from l100-l113 does not seem to make sense to me. IFNα could not decrease the activity of IDO by “stimulating production of IFNγ and TNFα” if these cytokines induce the IDO enzyme.
Materials and methods : l166 has a missing space.
Discussion : l261 “uptake” should be changed to “intake”.
L263 “comparing” should be changed to “compared”.
Author Response
Dear Sir or Madam,
We are very grateful for the review of our article titled “The role of the IFNγ rs2430561 polymorphism in inducible inflammation: contribution to metabolic syndrome in 45-60-year-old women”. We would like to thank you for all your comments and suggestions, which helped us to improve our manuscript. We are grateful for good words about the general value of the manuscript; it is very motivating for us.
The following corrections have been introduced in order to address the suggestions of the Reviewer 1 (marked in the manuscript with green):
1. There are some minor revisions to the writing style and quality of English usage that would improve clarity. For example, the authors should refine their use of the definite article. Moreover, there are a great many abbreviations used, not all of which are appropriately defined.
The text has been revised in terms of the language.
2. Abstract: l19 “whose production” should be changed to “the production of which”. L292 of the Discussion requires a similar revision.
Corrected.
3. Introduction: L37 “causes” should be changed to “means”. Paragraph from l100-l113 does not seem to make sense to me. IFNα could not decrease the activity of IDO by “stimulating production of IFNγ and TNFα” if these cytokines induce the IDO enzyme.
In order to maintain the clarity of the manuscript, we have decided to remove the sentence: “Interferon-alpha has a much weaker direct influence on IDO, but it can decrease its activity through stimulating the production of IFNγ and TNF-alpha”.
4. Materials and methods : l166 has a missing space.
Corrected.
5. Discussion : l261 “uptake” should be changed to “intake”. L263 “comparing” should be changed to “compared”.
Corrected.
We are very grateful for giving us the possibility of improving our manuscript. The article has been corrected according to the Reviewers’ suggestions.
Kindest regards
Małgorzata Szkup
Reviewer 4 Report
Manuscript entitled „The role of the IFNy rs2430561 polymorphism in inducible inflammation: contribution to metabolic syndrome in 45-60-year-old women” presents data from cross-sectional study, in which the association between polymorphic variants of IFNy and metabolic syndrome among females was analyzed. The authors analyzed also the association between IFNy polymorphism and serum levels of IFNy-inducible inflammatory cascade, which is supposed to be involved in the metabolic syndrome. Generally, the authors failed to indicate significant SNP effects. Since basically no significant SNP effect was observed (apart from minor differences in the kynurenine levels between heterozygotes with and without metabolic syndrome), the manuscript should be shortened because it is too long. Overall the manuscript is very similar to the previously published paper by these authors (Aging 10:72-82, 2018), in which the association between TNF-alpha polymorphism and inflammatory markers in the same group of subjects was analyzed. Furthermore, some of the data (metabolic markers) have been already published which is rather not accepted. Thus I do not recommend to publish the paper in the present form. The authors should revise the manuscript in order to show only new data.
Major comments
1. The title should be revised in my opinion since it suggests positive association between IFN rs2430561 and metabolic syndrome whereas no such association was found.
2. Introduction – I would suggest to include the information what is the metabolic syndrome.
3. Please add description of the statistical analysis method under each table.
Minor comments:
4. Paragraph “TRY metabolism pathways” – line 76, TDO should be in italics. In the last paragraph (page 3, line 116) – the study focuses on one polymorphism so single form should be used (polymorphism instead of polymorphisms). The same for the title of Table 3.
5. Materials and methods, paragraph 2.2, line 173 - please check the formatting.
6. Authors use two terms: kinerunine and kynerunine throughout the text. Please unify.
Author Response
Dear Sir or Madam,
We are very grateful for the review of our article titled “The role of the IFNγ rs2430561 polymorphism in inducible inflammation: contribution to metabolic syndrome in 45-60-year-old women”. We would like to thank you for all your comments and suggestions, which helped us to improve our manuscript.
The following corrections have been introduced in order to address the suggestions of the Reviewer 1 (marked in the manuscript with pink):
1. Manuscript entitled „The role of the IFNy rs2430561 polymorphism in inducible inflammation: contribution to metabolic syndrome in 45-60-year-old women” presents data from cross-sectional study, in which the association between polymorphic variants of IFNy and metabolic syndrome among females was analyzed. The authors analyzed also the association between IFNy polymorphism and serum levels of IFNy-inducible inflammatory cascade, which is supposed to be involved in the metabolic syndrome. Generally, the authors failed to indicate significant SNP effects. Since basically no significant SNP effect was observed (apart from minor differences in the kynurenine levels between heterozygotes with and without metabolic syndrome), the manuscript should be shortened because it is too long. Overall the manuscript is very similar to the previously published paper by these authors (Aging 10:72-82, 2018), in which the association between TNF-alpha polymorphism and inflammatory markers in the same group of subjects was analyzed. Furthermore, some of the data (metabolic markers) have been already published which is rather not accepted. Thus I do not recommend to publish the paper in the present form. The authors should revise the manuscript in order to show only new data.
Despite the fact that we have failed to change significant SNP effects, the information contained in the article is considered important for understanding the concepts of research and it would be difficult to give up their presentation.
It's true that we have previously published an article regarding the influence of TNF-alpha rs1800629 polymorphism on some inflammatory biomarkers, but we have described it thoroughly in the Materials and Methods section: “The research results presented in this manuscript are a part of a larger project, based on the same group, titled “The search for factors contributing to the metabolic syndrome among women aged 45-60 years” [31]”. The manuscript published in Aging describes another aspects of the same larger project based on the analysis of the same group of women. In order to maintain the clarity of the message in the current manuscript, it was necessary to present the characteristics of the study sample, which allowed the reader to get acquainted with the general health situation in the study sample, but it was not a clue of the topic. We want to emphasize that the current manuscript and the one published earlier in Aging are two independent realizations of research goals.
2. The title should be revised in my opinion since it suggests positive association between IFN rs2430561 and metabolic syndrome whereas no such association was found.
We have decided to changed the title to: “Searching for the role of IFNγ polymorphism rs2430561 in inducible inflammation: participation in the metabolic syndrome in 45-60-year-old women.”
2. Introduction – I would suggest to include the information what is the metabolic syndrome.
Thank you, we missed it. We have completed the introduction by adding the sentence: “Metabolic syndrome (MetS) is defined as a cluster of metabolic abnormalities, thus enhancing the risk of cardiovascular disease and premature death.”
3. Please add description of the statistical analysis method under each table.
A detailed description of all methods employed in the study has been included in the Materials and Methods section. We have not met with the practice of rewriting these methods under tables or figures. We believe that the standard method of describing statistical methods used by us is clear and should be clear enough for readers.
4. Paragraph “TRY metabolism pathways” – line 76, TDO should be in italics. In the last paragraph (page 3, line 116) – the study focuses on one polymorphism so single form should be used (polymorphism instead of polymorphisms). The same for the title of Table 3.
Thank you, we have corrected it.
5. Materials and methods, paragraph 2.2, line 173 - please check the formatting.
Thank you, we have corrected it.
6. Authors use two terms: kinerunine and kynerunine throughout the text. Please unify.
Thank you, we have unified it. “Kynerunine” is use throughout the text.
We are very grateful for giving us the possibility of improving our manuscript. The article has been corrected according to the Reviewers’ suggestions.
Kindest regards,
Małgorzata Szkup
Round 2
Reviewer 2 Report
The authors performed 5 times tests in Table 2, and 15 times tests in Table 4. This is multiple comparison, and leads Type I error. In Table 4, Kynurenine [ng/ml] is associated with Met S only in the A/T group; it does not shown an additive (dose-response) effect. The Bonferoni correction should be used. Moreover, this analysis method could not elucidate allele (addive, dominant, recessive) effects.
Author Response
Dear Sir or Madam,
We are very grateful for the review – round 2 of our article titled “The role of the IFNγ rs2430561 polymorphism in inducible inflammation: contribution to metabolic syndrome in 45-60-year-old women”. We have made every effort to clarify the contentious issues so that accepting our manuscript would be possible.
1. The authors performed 5 times tests in Table 2, adn 15 times tests in Table 4. This is multiple comparison, and leads Type I error. In Table 4, Kynurenine [ng/ml] is associated with Met S only in the A/T group; it does not shown an additive (dose-response) effect. The Bonferoni correction should be used. Moreover, this analysis method could not elucidate allele (addive, dominant, recessive) effects.
We appreciated the review point of view. However in the table 1 each inflammation marker is measured independently to another. We may use a multiple correction (there are many of them, not only Bonferroni one – which has moderate sensitivity and power; such as Tukey or Beniamin-Hohberg idea) when we are comparing the same variable between more than two groups, as the p value rise with following comparisons. Therefore we are really not convinced that in our case such multiple corrections should be done (and is necessary) as we compare only two groups (Met + and Met -).
Kindest regards,
Małgorzata Szkup
Reviewer 4 Report
I appreciate the authors' efforts to improve their manuscript and thank you for the response to my comments. Unfortunately there are still two points which should be revised in my opinion.
1. Statistics – what I meant was to add the name of statistical test/model under each table (not the description of statistical method). It would be easier for the reader, especially when many different tests were done. Besides IT IS very common and higly recommended to add such information under table data. I asked for that because I do not fully understand how did you analyse data in Table 4 „Analysis of the relationships between the IFNγ gene rs2430561 polymorphism and the levels of IFNγ-inducible inflammatory cascade markers with regard to MetS.” According to the description you provided in the text it seems that you conducted QTL analysis based on general linear model (in which I believe one quantitative data = inflammatory marker and two qualitative data (type of group and genotype) were included. Such model would generate information about the effect of: 1) group and 2) genotype; and after using post hoc analysis we can analyse differences between specific groups. So I wonder how did you indicate significant difference in kynurenine concentration between MetS+ and MetS- heterozygotes? Please provide more detailed description of this analysis and the name of the appropriate test/model under the table because in the current form it looks that you used simple Student’s t-test. Finally, I wonder whether it would not be more appropriate to use two-way ANOVA instead of QTL.
2. If you decided to show already published data such as metabolic markers, you should clearly state this and provide the reference wherever appropriate (page 5, from line 196). I agree that these data „allowed the reader to get acquainted with the general health situation in the study sample” but they are still part of the results and publishing the same results twice is defined as autoplagiarism. Furthermore, please not that there is no method description for analysis of these markers – you should also provide here reference to your previous paper.
Author Response
Dear Sir or Madam,
We are very grateful for the review of our article titled “The role of the IFNγ rs2430561 polymorphism in inducible inflammation: contribution to metabolic syndrome in 45-60-year-old women”. We have made every effort to clarify the contentious issues so that accepting our manuscript would be possible.
The following corrections have been introduced in order to address the suggestions of the Reviewer 1 (marked in the manuscript with red color):
1. Statistics – what I meant was to add the name of statistical test/model under each table (not the description of statistical method). It would be easier for the reader, especially when many different tests were done. Besides IT IS very common and higly recommended to add such information under table data. I asked for that because I do not fully understand how did you analyse data in Table 4 „Analysis of the relationships between the IFNγ gene rs2430561 polymorphism and the levels of IFNγ-inducible inflammatory cascade markers with regard to MetS.” According to the description you provided in the text it seems that you conducted QTL analysis based on general linear model (in which I believe one quantitative data = inflammatory marker and two qualitative data (type of group and genotype) were included. Such model would generate information about the effect of: 1) group and 2) genotype; and after using post hoc analysis we can analyse differences between specific groups. So I wonder how did you indicate significant difference in kynurenine concentration between MetS+ and MetS- heterozygotes? Please provide more detailed description of this analysis and the name of the appropriate test/model under the table because in the current form it looks that you used simple Student’s t-test. Finally, I wonder whether it would not be more appropriate to use two-way ANOVA instead of QTL.
We have add statistical test name under each table. We have also add some QTL explanation in the statistical description: “Co-dominant model was used and the model yields sample size, mean and standard error for each genotype, as well as mean difference and its 95% confidence interval [33]”. Additionally, we added one reference: Gonzalez, J.R.; Armengol, L.; Sole, X.; Guino, E.; Mercader, J.M.; Estivill, X.; Moreno, V. SNPassoc: an R package to perform whole genome association studies. Bioinformatics, 2007, 23(5), 654-5; DOI: 10.1093/bioinformatics/btm025” and we have revised the list of the references.
2. I you decided to show already published data such as metabolic markers, you should clearly state this and provide the reference wherever appropriate (page 5, from line 196). I agree that these data „allowed the reader to get acquainted with the general health situation in the study sample” but they are still part of the results and publishing the same results twice is defined as autoplagiarism. Furthermore, please not that there is no method description for analysis of these markers – you should also provide here reference to your previous paper.
We decided correct the suggested part of manuscript to exclude any suspicions of autoplagiarism. We have removed table 1 and part of the description regarding metabolic markers. In addition, we have added information where this data can be found by readers:
We compared the women qualified for subgroups MetS+ and MetS-. All parameters statistically significantly differed between both groups (p < 0.001). Particular attention should be paid to the fact that even in the MetS- group, the average values for one of the five symptoms – waist size - analyzed exceeded the values considered by the IDF as correct. The average waist size in the MetS+ group was 93.3±11 cm (compared to 85.4±11.2 cm in the MetS- group). The mean values for three out of five symptoms in both groups were within normal ranges: TG level (137.6 (102.0 – 189.8) mg / dl v. 84.8 (65.0 – 112.1) mg / dl, MetS+ v. MetS- respectively), HDL level (56.5±16.8 mg / dl v. 70.0±16.0 mg / dl, MetS+ v. MetS- respectively), and diastolic blood pressure (83.9±9.4 mmHg v. 75.9±9.7 mmHg MetS+ v. MetS- respectively). In the MetS+ group, the average values of other MetS parameters were elevated, while in the MetS- group they were normal: fasting glycemia (100.8 mg / dl (86.9 – 119.0) v. 83.2 (77.4 – 90.7) mg / dl, MetS+ v. MetS- respectively) and systolic blood pressure (137.2 ± 15.3 mmHg v. 119.1 ± 14.8 mmHg, MetS+ v. MetS- respectively). Table 1 shows the number and percentage of the participants in the MetS+ and MetS- groups with values exceeding normal ranges. The most common problems in both groups were hypertension and visceral obesity (Table 1). Acurate characteristics of the study sample with regard to division into MetS+ and MetS- groups, according to IDF diagnostic criteria from 2009 is available at Szkup et al. because the research results presented in this manuscript are a part of a larger project, based on the same group [31].
Table 1. Characteristics of the study sample with regard to division into MetS+ and MetS- groups, according to IDF diagnostic criteria from 2009.
MetS components | Reference range | MetS+ above the norm [N(%)] (N = 118) | MetS- above the norm [N(%)] (N = 298) |
Waist size [cm]* | ≤ 80 | 97 (82.2) | 62 (20.8) |
Hyperglycemia [mg/dl] * | ≤100 | 60 (50.8) | 14 (4.7) |
TG [mg/dl] * | < 150 | 75 (63.6) | 32 (10.7) |
HDL [mg/dl] * | > 50 | 61 (51.7) | 21 (7.1) |
Hypertension [mmHg] * | sBP ≤ 130 or/and dBP ≤ 85 | 97 (82.2) | 62 (20.8) |
MetS+ - group included women who met the criteria for metabolic syndrome according to the IDF diagnostic criteria from 2009; MetS- - women without MetS; N – number of cases; TG – triglycerides; HDL - high-density lipoprotein; * statistically significant p <0.001< strong=""> |
In addition, we've added location information for more information about the method description for analysis of metabolic markers: The research results presented in this manuscript are a part of a larger project, based on the same group, titled “The search for factors contributing to the metabolic syndrome among women aged 45-60 years”, in which we also provided more detailed information about the description of the research procedure [31].
Kindest regards,
Małgorzata Szkup